# Computational Modeling of Uncertainty and Volatility Beliefs in Escape-Avoidance Learning: Comparing Individuals with and Without Suicidal Ideation

**DOI:** 10.3390/jpm15120604

**Published:** 2025-12-05

**Authors:** Miguel Blacutt, Caitlin M. O’Loughlin, Brooke A. Ammerman

**Affiliations:** 1Department of Psychology, University of Notre Dame, Corbett Family Hall, 18931 Moose Krause Cir, Notre Dame, IN 46556, USA; 2Corporal Michael J. Crescenz VA Medical Center, 3900 Woodland Ave, Philadelphia, PA 19104, USA; caitlin.oloughlin@va.gov; 3Department of Psychology, University of Wisconsin—Madison, 1202 W Johnson St, Madison, WI 53706, USA; baammerman@wisc.edu

**Keywords:** suicide, suicidal ideation, computational, belief, uncertainty, decision making, cognitive, escape, reinforcement learning, hierarchical gaussian filter

## Abstract

**Background/Objectives**: Computational studies using drift diffusion models on go/no-go escape tasks consistently show that individuals with suicidal ideation (SI) preferentially engage in active escape from negative emotional states. This study extends these findings by examining how individuals with SI update beliefs about action–outcome contingencies and uncertainty when trying to escape an aversive state. **Methods**: Undergraduate students with (*n* = 58) and without (*n* = 62) a lifetime history of SI made active (go) or passive (no-go) choices in response to stimuli to escape or avoid an unpleasant state in a laboratory-based negative reinforcement task. A Hierarchical Gaussian Filter (HGF) was used to estimate trial-by-trial trajectories of contingency and volatility beliefs, along with their uncertainties, prediction errors (precision-weighted), and dynamic learning rates, as well as fixed parameters at the person level. Bayesian mixed-effects models were used to examine the relationship between trial number, SI history, trial type, and all two-way interactions on HGF parameters. **Results**: We did not find an effect of SI history, trial type, or their interactions on perceived volatility of reward contingencies. At the trial level, however, participants with a history of SI developed progressively stronger contingency beliefs while simultaneously perceiving the environment as increasingly stable compared to those without SI experiences. Despite this rigidity, they maintained higher uncertainty during escape trials. Participants with an SI history had higher dynamic learning rates during escape trials compared to those without SI experiences. **Conclusions**: Individuals with an SI history showed a combination of cognitive inflexibility and hyper-reactivity to prediction errors in escape-related contexts. This combination may help explain difficulties in adapting to changing environments and in regulating responses to stress, both of which are relevant for suicide risk.

## 1. Introduction

Suicide is a leading cause of death globally, claiming approximately 800,000 deaths each year and representing the second leading cause of death in people aged 10–34 and the fifth leading cause of death in people aged 35–54 in the United States [1]. These rates have been increasing since 1999 and currently generate a national annual cost of approximately USD 100 billion [2]. Despite the magnitude of this public health crisis, our ability to predict and prevent suicidal thoughts and behaviors remains remarkably limited. Traditional approaches have emphasized risk factors and correlates of suicidal ideation (SI), but far less is known about the cognitive mechanisms through which individuals with SI histories process information and make decisions, particularly when attempting to escape aversive internal states. Computational psychiatry offers a promising avenue for clarifying these processes, moving beyond descriptive accounts and toward mechanistic explanations of suicidal cognition.

Several contemporary theories of suicide, such as the Escape Theory of Suicide [3], the Integrated Motivational–Volitional Model [4], and the Interpersonal Theory [5], converge on the idea that SI is driven by a desire to escape persistent negative states. Complementing these theories, three computational studies have examined escape preferences using a paradigm where participants learn whether to press a key (go; active) or withhold pressing (no-go; passive) in response to fractal images (see Section 2.2 for an illustration of this task) to either escape an aversive sound (i.e., trial starts with sound and a correct response terminates it) or avoid the sound (i.e., trial starts with silence and a correct response prevents its onset). 

Each of these studies used reinforcement learning drift diffusion models to decompose decision-making processes into specific cognitive mechanisms. These models conceptualize decisions as evidence accumulation processes, where information gradually builds toward either a go or no-go response threshold, with the starting point of this accumulation indexing an escape bias. See Figure 1 for a visual depiction of how starting-point bias shapes the accumulation process and guides go versus no-go decisions. Millner et al. found that veterans with prior histories of suicidal thoughts and behaviors showed significantly stronger escape biases compared to psychiatric controls [6]. Blacutt et al. replicated this finding in an undergraduate sample and demonstrated that the escape bias remained among those with a lifetime history SI after controlling for impulsivity facets [7]. Jaroszewski et al. extended this work by introducing suicide-specific stimuli, where fractal images were either suicide-related (e.g., an image of a gun or a knife facing the participant) or positive (e.g., a candy bar facing the participant) [8]. These authors found that those with recent SI showed stronger escape biases when responding to suicide-related stimuli. Together, these studies indicate that SI is associated with escape-related preferences, supporting theories that view escape as a motivation for SI.

A limitation of these prior studies is that the drift diffusion approach utilized assumes that the direction and rate of the decision-making process are solely driven by the latent value of each choice given a particular stimulus [9,10], without accounting for the uncertainty about these values. This limitation is particularly relevant for understanding suicide risk, where decision-making is rarely driven by perceived value alone (i.e., the perceived relative rewards and costs of living and dying) [11,12]. In fact, periods of elevated SI are often characterized by intense ambivalence, where individuals may experience the desire to escape pain while remaining uncertain about how much they value continuing to live and whether death is what they truly want [13,14]. This uncertainty also extends to adaptive behaviors: when individuals lack confidence that coping strategies will provide relief, they may default to maladaptive escape-oriented behaviors either due to greater perceived certainty that death will end emotional pain or, paradoxically, due to global uncertainty that prevents them from ruling out death as a viable solution [11]. Furthermore, rigid certainty in negative beliefs, such as hopelessness (“nothing will ever help”), can entrench SI [5,14,15], discourage the use of coping skills, and prevent adaptive learning. By examining not just what individuals value but also how certain they that their actions will lead to outcomes they value and whether they expect the environment to change, we can better capture the cognitive dynamics underlying SI.

A Hierarchical Gaussian Filter (HGF) approach can address these gaps by explicitly modeling overall and trial-level belief formation and uncertainty using a coupled hierarchy of Gaussian random walks [16,17]. Unlike drift diffusion models, the HGF approach can simultaneously track experienced action–outcome pairings, referred to as “contingencies” (level 1), beliefs and uncertainty about these contingencies (level 2), and beliefs and uncertainty about the volatility of the environment or how quickly those contingencies are expected to change (level 3). Put simply, contingency beliefs reflect what individuals expect will happen when they take a specific action; uncertainty reflects how confident they are in that expectation; and volatility reflects whether they believe these learned contingencies are stable or shifting. For example, within the Interpersonal Theory framework [5], a person who has experienced SI and feels chronically disconnected might believe that positive social interactions can relieve distress (a contingency belief) but be uncertain about whether such outcomes will actually occur (uncertainty), while also believing that their social circumstances rarely change (low perceived volatility). Together, the combination of strong yet uncertain contingency beliefs and low volatility expectations can reinforce hopelessness and rigid, escape-oriented thinking.

### Study Aims

The present study employs an HGF to explore how individuals with and without a history of SI differ in their contingency beliefs, contingency uncertainty, and volatility beliefs during escape and avoidance learning. Building on prior work demonstrating escape biases in SI [6,7,8] and theoretical accounts of ambivalence and uncertainty during suicidal crises [12,13], we hypothesized that participants with SI histories would develop progressively stronger contingency beliefs across the task and maintain higher contingency uncertainty in escape trials compared to those without a history of SI. Given the novelty of dynamic learning rates and prediction-weighted prediction errors in the SI literature, analyses about these parameters remained exploratory. This approach extends prior reinforcement learning research by directly modeling how uncertainty and perceived environmental stability interact with escape and avoidance processes in those with SI, addressing a key gap in prior work that has focused primarily on value-based decision biases without considering uncertainty around them.

## 2. Materials and Methods

### 2.1. Participants and Procedures

Participants were recruited through SONA Systems, a secure, web-based platform used to recruit undergraduate participants for research studies completed in exchange for a course credit. This represented a convenience sample drawn from the University of Notre Dame’s research participation pool. Participants with and without a lifetime SI history were included; the only exclusion criteria were reporting high levels of current depressive symptomatology (i.e., score of 20 or greater on the Patient Health Questionnaire-9) [18] or current suicidality (i.e., reporting SI in the past week) due to ethical considerations, as the larger study included negative mood induction. These criteria were assessed via self-reported responses on the PHQ-9 prior to the laboratory session. All participants provided informed consent before participating in this study. Data were de-identified via study-specific codes and were stored on an institutional secure internal server for sensitive data. Participants were compensated via SONA credit (1 credit/hour) and were assessed for suicide risk at the beginning and end of the laboratory session using the University of Washington Risk Assessment Protocol [19]. Either a licensed clinical psychologist or a doctoral-level trained graduate research assistant was available for risk management in the event of high suicide risk (i.e., >4/7 intent to act on urges and an absence of a safety plan); no participants demonstrated elevated suicide risk as per the risk criteria suggested by Linehan et al. [19]. The study was carried out in accordance with the Declaration of Helsinki and was approved by the University of Notre Dame’s IRB.

### 2.2. Measures

#### 2.2.1. Clinical Interview

Suicidal Ideation. Lifetime SI (i.e., “Have you ever had thoughts of killing yourself?” [no/yes]) was assessed as part of a semi-structured interview, the Self-Injurious Thoughts and Behaviors Interview [20]. Response to this measure was used to generate two groups: one with a lifetime SI history and one without this history.

#### 2.2.2. Behavioral Task

We used a negative reinforcement learning task devised to assess avoid and escape behavior (see Millner et al., [21], for a thorough description of this task), which was administered on a desktop computer in-laboratory under the supervision of a trained study team member. The task involves two conditions: escape and avoidance. During escape trials, participants were initially presented with an aversive sound and had to correctly respond to a visual cue (a fractal image) to terminate the sound. During avoidance trials, participants were presented with silence and had to correctly respond to a cue to prevent the aversive sound from starting. The fractal images consistently signaled whether participants should press a button (go; active) or withhold pressing (no-go; passive) to achieve (i.e., during escape trials) or maintain silence (i.e., during avoidance trials). Thus, the task yields four response types: (1) active escape, requiring a locomotor action to cease the sound; (2) passive escape, requiring inaction to cease the sound; (3) active avoidance, requiring a locomotor action to prevent the sound; and (4) passive avoidance, requiring inaction to prevent the sound (see Table 1). 

The aversive sound was the sound of a fork scraping on a slate, altered with a high-frequency sound, presented over headphones at 80–85 dB. Each cue was presented for 1 s, during which participants could not respond, followed by a 2 s window where they could respond at any time. The aversive sound was presented for 2 s after incorrect choices (e.g., negative feedback), and silence was presented for 750 ms after correct choices (e.g., positive feedback); after receiving the aversive sound or silence, a 1 s inter-trial interval with no sound and a screen displaying a white cross was presented. Before starting the task, participants received brief instructions stating that they would need to choose to either click or not click their mouse in response to images to either cease or prevent a loud and aversive sound. Further, participants were instructed to try to learn what the best response is for each image. Participants completed a total of 120 trials; no participant discontinued the experiment. An illustration of successful responses in both escape and avoidance conditions can be seen in Figure 2.

#### 2.2.3. Computational Modeling

Overview of Hierarchical Gaussian Filter Model. To examine differences in contingency beliefs, contingency uncertainty, and volatility expectations between participants with and without a history of SI, we used an HGF model [16,17]. This model is a generative Bayesian model of learning that represents how individuals update their beliefs about the environment across multiple levels. At each level, beliefs are modeled as Gaussian distributions, with means representing expectations of action–outcome contingencies and variances representing uncertainty about these contingencies. Higher levels track the mean and volatility beliefs of the levels below. In the context of our task, this framework allows us to formally characterize how participants learn whether their actions (go vs. no-go) control aversive sounds (whether in escape or avoidance trials) and whether such contingencies are perceived as stable or changing. To estimate participants’ trial-by-trial beliefs, we fit the HGF to each individual’s task data separately using variational approximation, implemented in the HGF Toolbox version 7.1 [16,17,22] in MATLAB version 2025b with default initial parameters. Formally, the generative process is defined as follows:(1)$$x_{1}^{\left(t\right)}∼\mathrm{Bernoulli}\left(s\left(x_{2}^{\left(t\right)}\right)\right)$$(2)$$x_{2}^{\left(t\right)}∼N\left(x_{2}^{\left(t-1\right)},\;exp\;\left(x_{3}^{\left(t\right)}+\omega_{2}\right)\right)$$(3)$$x_{3}^{\left(t\right)}∼N\left(x_{3}^{\left(t-1\right)},\;exp\;\left(\omega_{3}\right)\right)$$
where ω_2_ captures the tonic evolution rate at level 2 (the fixed tendency to expect contingency changes), while ω_3_ reflects the tonic evolution rate at level 3 (the fixed tendency to expect volatility itself to fluctuate).

At each level *i*, the model maintains both a posterior mean belief *μ_i_*^(*t*)^ and a posterior variance σ_i_^2(t)^, where variance quantifies uncertainty about the state. Belief updating is driven by a precision-weighted prediction error:(4)$$\mu_{i}^{\left(t\right)}=\mu_{i}^{\left(t-1\right)}+\psi_{i}^{\left(t\right)}\delta_{i-1}^{\left(t\right)}$$(5)$$\sigma_{i}^{2\left(t\right)}=f\left(\sigma_{i}^{2\left(t-1\right)},\pi_{i}^{\left(t\right)},{\hat{\pi }}_{i-1}^{\left(t\right)}\right)$$
where *δ_i−*1*_*^(*t*)^ is the raw prediction error from the level below, and *ψ_i_*^(*t*)^ is the dynamic learning rate defined as the ratio of posterior precision estimates:(6)$$\psi_{i}^{\left(t\right)}=\frac{{\hat{\pi }}_{i-1}^{\left(t\right)}}{\pi_{i}^{\left(t\right)}}$$
where *π_i_* and *π_i*−1*_* are the posterior precisions at level *i* and the preceding level, respectively.

Interpreting Trial-Level Estimates. In the context of our escape/avoidance go/no-go task, these terms provide concrete insight into learning and decision-making among those with SI. At the second level, *μ_*2*_*^(*t*)^ reflects the participant’s belief about the probability that a given action–state pairing, such as pressing “go” on an escape trial, will successfully lead to silence, while *σ_*2*_*^(*t*)^ captures the uncertainty about this contingency. At the third level, *μ_*3*_*^(*t*)^ encodes beliefs about the volatility of contingencies, reflecting whether the effectiveness of “go” versus “no-go” is stable or changing across trials, while *σ_*3*_*^(*t*)^ reflects uncertainty about that volatility. Belief updating is driven by mismatches between predicted and experienced task outcomes (prediction errors), which are scaled by dynamic learning rates *ψ_*2*_*^(*t*)^ and *ψ_*3*_*^(*t*)^ that determine the strength of these errors in influencing updates at each level. The product of raw prediction errors with dynamic learning rates yields precision-weighted prediction errors *ε_*2*_*^(*t*)^ and *ε_*3*_*^(*t*)^, which are the learning signals that update contingency and volatility beliefs. Finally, a response model links beliefs to choices: the inverse temperature parameter *β* governs how consistently participants act in line with their current contingency belief, with higher *β* producing more deterministic choices and lower β producing noisier responses.

#### 2.2.4. Statistical Analysis

Bayesian mixed-effects models were used to examine both fixed HGF parameters and trial-level trajectories. For fixed effects, separate models tested the influence of SI history, condition (escape vs. avoidance), and their interaction on perceptions of tonic volatility and choice stochasticity, with a random intercept per participant. For trial-level dynamics, we fit models including trial number, SI history, condition, and all two-way interactions as predictors of HGF-derived trajectories of contingency beliefs, volatility beliefs, respective uncertainties, precision-weighted prediction errors, and dynamic learning rates. All models were implemented in R version 4.4.1 using the brms package version 2.23.0 [23], with four chains, 1000 warm-up iterations, 4000 post-warmup iterations per chain, and a random intercept for each participant. Convergence was assessed using Rhat < 1.05 [24] and visual inspection of trace plots, and reliability of estimates was confirmed with effective sample sizes > 400. Estimates were considered significant if their 95% credible intervals excluded zero. All models used non-informative priors, and model assumptions were verified via visual inspection of residuals and random effects. Trial-by-trial trajectory modeling was employed to capture dynamic changes in belief updating across the task, as aggregate parameters would obscure temporal patterns of learning that may differ between groups with and without SI history. There was no missing trial-level data. As part of a larger study, participants underwent a mood induction protocol; however, there were no differences between groups on the current study’s primary outcomes (*p*’s = 0.36–0.94).

## 3. Results

The final sample consisted of 120 undergraduate students from a private Midwestern university who either had a history of SI (*n* = 58) or no history of SI (*n* = 62) and were invited to an in-laboratory session. Participants had a mean age of 19.1 ± 1.3 years and self-identified as Asian (10.8%), Black (8.3%), Middle Eastern (4.2%), Multiracial (8.3%), and White (68.3%); see Table 2. Participants with a history of SI were significantly younger than those without an SI history (standardized mean difference = −0.64).

### 3.1. Fixed Parameters

Bayesian mixed-effects models did not reveal an effect of SI history, condition, or their interaction on perceptions of volatility and choice stochasticity (Table 3).

### 3.2. Trial-by-Trial Parameters

All trial-level parameters can be found in Table 3.

Contingency and Volatility Beliefs. We found a trial number by SI history interaction on contingency beliefs. Relative to those without SI experiences, participants with an SI history developed progressively stronger contingency beliefs across the task (Figure 3). Moreover, we found a negative interaction of trial number by SI history on volatility beliefs, suggesting that participants with an SI history became increasingly expectant that the task environment would remain unchanged.

Contingency and Volatility Uncertainty. We found an escape trial by SI history interaction on contingency uncertainty. Participants with an SI history maintained higher contingency uncertainty in escape trials compared to those without SI experiences, meaning they remained less confident about which actions would successfully terminate the sound. This suggests difficulty consolidating reward contingencies when actively terminating an aversive state (Figure 3). Further, we found a negative trial number by SI history interaction on volatility uncertainty, indicating that participants with an SI history became more certain that the environment would not change throughout the task.

Precision-Weighted Prediction Errors. We did not find any effect of SI history on the contingency or volatility of precision-weighted prediction errors.

Dynamic Learning Rates. We found an escape trial by SI history interaction on the contingency learning rate, which suggests that participants with an SI history weighted surprising outcomes (e.g., those with high prediction errors) more strongly when updating their contingency beliefs within that same trial than those without SI experiences when actively trying to terminate the sound (Figure 3). In other words, those with SI histories adjusted their expectations by a greater magnitude in response to unexpected results. We did not find any effect of an SI history on the volatility learning rate.

## 4. Discussion

The present study used HGF to examine how individuals with and without SI histories differ in their contingency belief updating and uncertainty processing during escape and avoidance learning. We did not find significant differences in fixed estimates of volatility or stochastic decision-making between groups. However, we observed several notable findings in the trial-level data. First, aligned with our hypotheses, individuals with SI histories, relative to those without SI histories, showed greater increases in their belief that an active response would result in successfully controlling the sound as the task progressed (e.g., viewing the outcome as increasingly stable). Second, aligned with our hypotheses, as the task progressed, those with SI histories, compared to those without SI histories, developed less contingency uncertainty obtained from each choice (e.g., became more certain that “go” [i.e., escape] leads to successful control of the sound). Lastly, individuals with lifetime SI histories, relative to those without SI histories, developed less uncertainty about the volatility of reward contingencies throughout the task (e.g., perceived the environment as less subject to change).

These findings align with established cognitive and escape theories of SI [3,11,25]. Specifically, the findings that individuals with an SI history developed progressively stronger contingency beliefs while simultaneously showing decreased expectations of volatility, combined with elevated uncertainty during escape contexts, align with cognitive theories of SI. Dombrovski and Hallquist propose that SI emerges when impaired learning and cognitive constraints narrow the range of considered solutions, restrict the ability to simulate alternative futures, and bias choice toward seemingly simple but maladaptive options [11]. Although ecologically valid research will be necessary to establish whether these processes generalize beyond the laboratory, we propose that our results likely map directly onto this framework: stronger contingency beliefs may reflect a constricted solution space where escape responses dominate; lower volatility expectations may reflect the rigid assumption that the environment will not change, aligning with the previously established role of hopelessness in SI [5,14,26,27]; and persistent uncertainty about escape contingencies may reflect difficulty consolidating which actions reliably provide relief, leaving individuals simultaneously rigid in worldview yet uncertain in execution. Among those with SI, such rigidity may manifest as narrowed thinking, that is, myopically present-focused, leads to limited exploration of alternative solutions [3,12], and potentially heightens hopelessness over time [28], leaving suicide increasingly salient as an escape option. This interpretation converges with Rudd’s (2000) [25] description of “suicidal mode,” in which cognitive, affective, behavioral, and physiological elements align to narrow attention toward suicide as the most viable escape strategy. Within this mode, individuals form a suicidal belief system characterized by the growing certainty that they are unlovable and that their future will not change while struggling to integrate information that contradicts these assumptions. The emergence of a suicidal mode and accompanying belief system is consistent with the belief and uncertainty dynamics identified in the present study.

In addition to being aligned with modern theories of suicide, these findings may provide a mechanistic account for empirical findings in the related suicide literature. Al-Dajani et al. found that a stronger belief that suicide is a method of escape amplified the effect of low emotional clarity on SI at a six-month follow-up, suggesting that emotional uncertainty heightens the risk that escape-oriented beliefs sustain suicidal thinking [29]. Our finding of elevated dynamic learning rates during escape trials in individuals with SI history suggests a potential computational mechanism underlying this relationship. Specifically, heightened dynamic learning rates during escape trials relative to the control group indicate that individuals with SI may overweight prediction errors when attempting to terminate aversive states, potentially producing unstable action–outcome representations that manifest as both contingency uncertainty and broader emotional uncertainty. This decision-making instability may perpetuate SI by preventing the consolidation of reliable mental models for emotion regulation. In other words, unstable expectations about which strategies reduce distress leave individuals uncertain about what will help, undermining coping. Conversely, heightened dynamic learning rates may cause them to overvalue short-term relief from suicidal thoughts or behaviors, reinforcing suicide as an escape option. Thus, excessive updating during escape contexts may represent a computational pathway through which escape motivations and emotional uncertainty jointly maintain suicidal cognition.

Previous studies using the same escape task with drift diffusion modeling have shown that individuals with SI histories display a bias toward active escape, suggesting that they begin each escape trial predisposed to favor action (go) over inaction (no-go) [6,7,8]. The present findings provide a mechanistic account for this escape preference. Across the task, participants with SI histories demonstrated strengthened contingency beliefs in favor of active escape while concurrently showing reduced uncertainty about these contingencies and diminished expectations of environmental change. This pattern of rigid belief formation coupled with elevated uncertainty during escape trials suggests that escape preferences may persist because individuals both overvalue active escape and fail to update these beliefs when the environment changes.

The computational insights from the present study yield several potentially important clinical implications for treating individuals with SI. Given that those with SI generated more rigid contingency beliefs but retained uncertainty regarding alleviating ongoing distress, it could be particularly important for clinicians to focus on practicing concrete coping skills repeatedly until they become reliable alternatives to escape-seeking behaviors. Clinicians may choose to incorporate intensive in-session skill practice with frequent homework follow-up. Through consistent practice across varied emotional contexts, patients may be able to develop more precise internal models, of which the strategies provide relief in specific situations, shifting from rigid yet uncertain beliefs to flexible and confident coping repertoires. It may be beneficial for clinicians to ask clients to maintain written records of attempted emotion regulation strategies and their efficacy. This serves two purposes: First, it creates a period of reflection about coping skill efficacy, likely enhancing emotional clarity and strengthening reinforcement learning; second, given the heightened weighting of prediction errors during escape trials observed in our data, this tracking may help gradually calibrate expectations about strategy effectiveness. Given the non-clinical sample, these clinical implications should be considered preliminary theoretical applications that require empirical validation in clinical treatment settings before implementation.

Several limitations warrant consideration when interpreting these findings. First, the cross-sectional design precludes causal inferences about whether the observed computational differences represent vulnerability factors for SI development or consequences of having experienced SI, and the reliance on lifetime SI history rather than current ideation may obscure important differences between active and remitted states. Second, the sole recruitment of an undergraduate student sample with a relatively young age (mean 19.1 years) restricts our ability to determine whether these computational patterns persist across the lifespan. Third, the exclusion of individuals with severe current depression or recent SI limits generalizability to clinical populations, as our sample may not capture the diverse range of suicidal experiences seen in individuals with active suicidality or diagnosed mental health disorders. Fourth, we did not exclude participants based on hearing impairments, potentially introducing variability in auditory perception of the aversive sound. Fifth, to aid model identifiability, we fixed the coupling coefficient at 1. Although this is standard practice, it assumes that each participant’s estimate of volatility influences contingency learning in the same way across individuals and potentially masks person-specific differences in how volatility affects learning [16,17,22]. This assumption is conceptually similar to latent growth curve models where slope loadings are fixed across individuals, potentially obscuring heterogeneity in change processes even if slopes are generated at the person-specific level [30]. Future research with larger samples and more trials per participant could relax this assumption to capture individual variability in how volatility beliefs shape learning. Sixth, the task’s use of an aversive sound may not adequately capture the complexity of unpredictable, uncontrollable psychological pain experienced during real-world suicidal crises, limiting ecological validity. Seventh, factors not included in the statistical analysis, such as baseline anxiety, depression, and cognitive ability, may have influenced learning patterns and could represent potential confounders. Despite these limitations, this study has several strengths. First, it applies a sophisticated computational approach using the HGF that extends beyond previous drift diffusion models by decomposing multiple levels of belief formation and uncertainty processing, revealing temporal patterns of learning that simpler models would miss. Second, the use of trial-by-trial trajectory analysis allowed us to capture dynamic changes in contingency and volatility belief and uncertainty updating across the task, revealing how individuals with SI histories develop increasingly rigid beliefs about environmental stability while maintaining elevated uncertainty specifically during escape contexts, a pattern that would be obscured by examining only the aggregate performance or fixed parameters. Third, the integration of escape and avoidance conditions within the same paradigm enables direct comparison of learning processes across motivationally distinct contexts, identifying escape-specific computational signatures that provide mechanistic specificity directly informing theoretical models of SI.

## 5. Conclusions

The present study advances the understanding of SI by applying a computational lens that captures the dynamic interplay between belief formation, uncertainty processing, and escape motivation. The findings from the HGF model suggest that individuals with a history of SI develop increasingly rigid beliefs about environmental stability while simultaneously showing escape-specific persistence of contingency uncertainty, even as their uncertainty about volatility decreased across the task. These findings extend prior drift diffusion work by identifying not only an escape bias but also the impaired learning mechanisms that likely sustain it, potentially offering mechanistic insight into why suicidal thinking persists despite repeated ineffective escape attempts. These results suggest it may be prudent for clinicians treating individuals with SI to emphasize strengthening precise, reliable coping strategies and collaboratively calibrating expectations about the effectiveness of these techniques. Future research should test whether these computational markers prospectively predict the onset, maintenance, or escalation of suicidal thinking in high-risk populations and how they respond to specific treatment, thereby clarifying their potential as targets for personalized intervention. Moreover, future studies should examine how belief-updating dynamics relate to broader cognitive and emotional processes such as regulation, flexibility, and impulsivity.

## Figures and Tables

**Figure 1 jpm-15-00604-f001:**
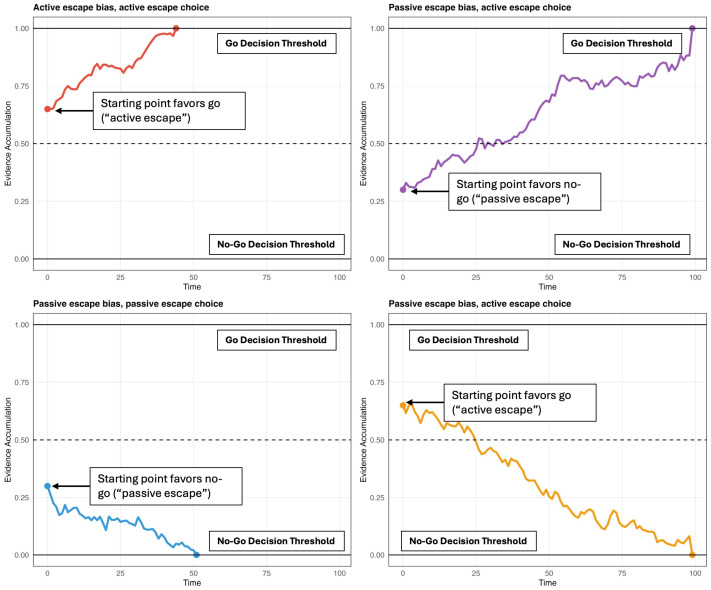
Drift diffusion decision-making process in a go/no-go escape-avoidance task. Quadrants illustrate combinations of starting points (active vs. passive escape bias) and decision outcomes (go vs. no-go), with the go and no-go boundaries corresponding to the decision thresholds. Higher starting points reflect active escape bias, whereas lower starting points reflect passive escape bias. Evidence trajectories terminate when the accumulation process reaches either the go threshold (1.0 on the y-axis) or the no-go threshold (0.0 on the y-axis). Note. Time on the x-axis reflects model time steps and is expressed in arbitrary units.

**Figure 2 jpm-15-00604-f002:**
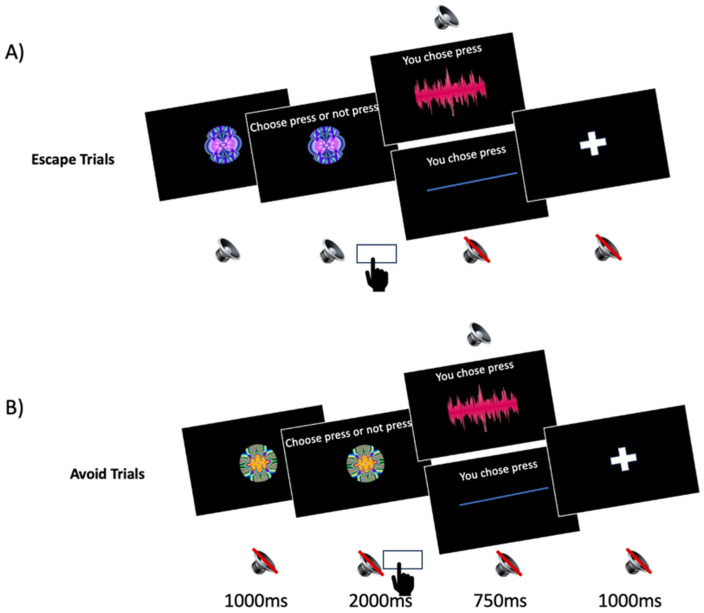
Example of successful go tasks for both escape and avoid conditions. (**A**) The participant is shown an aversive sound and a stimulus indicating they need to press a button to stop it. The participant presses the button correctly, and the noise stops. (**B**) The participant hears silence and a stimulus indicating they need to press a button to prevent the sound from starting. The participant presses the button correctly, and the sound does not begin.

**Figure 3 jpm-15-00604-f003:**
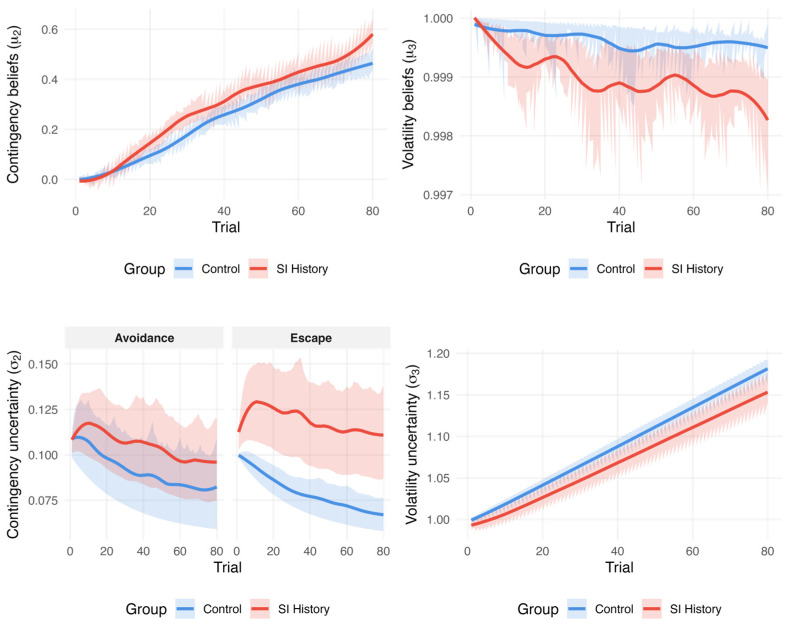
Trial-by-trial trajectories of Hierarchical Gaussian Filter parameters comparing individuals with lifetime suicidal ideation (SI) history and without it. Note: *μ_*2*_* = contingency beliefs; *μ*_3_ = volatility beliefs; *σ_*2*_* = contingency uncertainty; *σ_*3*_* = volatility uncertainty; *ψ_*2*_* = precision weight at level 2; and *ψ_*3*_* = precision weight at level 3. Precision-weighted prediction errors (*ε_*2*_* and *ε_*3*_*) were not depicted, as there were no significant effects of SI history. Contingency uncertainty separates avoidance and escape to showcase interaction.

**Table 1 jpm-15-00604-t001:** Task conditions, types of responses, and escape/avoidance behaviors they map onto.

Trial Description	Correct Response	
	Go	No-Go
Trial start: Aversive sound Correct response: Terminates sound Incorrect response: Sound continues	Active escape	Passive escape
Trial start: Silence Correct response: Silence continues Incorrect response: Sound starts	Active avoidance	Passive avoidance

**Table 2 jpm-15-00604-t002:** Participant demographics, task accuracy, and group comparisons.

	SI History (*n* = 58)	No SI History (*n* = 62)	Comparison
Age	18.7 ± 0.88	19.5 ± 1.54	*t*(86.5) = −3.3, *p* = 0.001 ^a^
Sex (Female), %(*n*)	65.0%	66.1%	*χ*^2^(1) < 0.0001, *p* = 0.99 ^b^
Race (Non-White), %(*n*)			*χ*^2^(1) = 1.51, *p* = 0.21 ^b^
Asian	12.1% (7)	9.7% (6)	–
Black	13.8% (8)	3.2% (2)	–
Middle Eastern	5.2% (3)	3.2% (2)	–
Multiracial	6.9% (4)	9.7% (6)	–
White	62.1% (36)	74.2% (46)	–
Accuracy (%)			–
Escape Trials	77.7 ± 17.3	80.4 ± 15.8	*t*(115.9) = −0.67 *p* = 0.51 ^a^
Avoid Trials	75.6 ± 16.2	78.5 ± 17.3	*t*(117.9) = −0.63, *p* = 0.53 ^a^
Go Trials	79.6 ± 19.0	78.5 ± 18.7	*t*(110.1) = 0.10, *p* = 0.91 ^a^
No-Go Trials	74.2 ± 20.3	79.9 ± 15.5	*t*(118.0) = −1.21, *p* = 0.22 ^a^

Note: ^a^ Comparison performed with independent sample *t*-tests. ^b^ Comparisons conducted with chi-square test of independence.

**Table 3 jpm-15-00604-t003:** Posterior means and credible intervals from Bayesian mixed-effects models examining the effects of suicidal ideation history, escape/avoidance condition, and trial number on Hierarchical Gaussian Filter learning parameters.

	Intercept	Trial #	SI History	Condition	Trial # *SI History	Trial # *Escape	SI History *Escape
** *ω* _2_ **	**−8.39 [−8.74, −8.04]**	-	0.09 [−0.42, 0.58]	−0.13 [−0.61, 0.35]	-	-	0.45 [−0.24, 1.14]
** *ω* _3_ **	**−5.99 [−6.00, −5.98]**	-	−0.00 [−0.02, 0.01]	0.01 [−0.01, 0.03]	-	-	−0.00 [−0.03, 0.02]
** *β* **	**1.70 [0.06, 3.31]**	-	0.80 [−1.62, 3.18]	**1.10 [0.11, 2.09]**	-	-	−0.56 [−2.00, 0.90]
** *μ* _2_ **	−0.0199 [−0.0745, 0.0322]	**0.00660 [0.00636, 0.00684]**	0.0179 [−0.0539, 0.0958]	0.0152 [−0.001, 0.0312]	**0.00113 [0.00077, 0.00149]**	**−0.00082 [−0.00115, −0.00049]**	−0.0188 [−0.0425, 0.00491]
** *μ* _3_ **	**1.00 [1.00, 1.00]**	−0.00 [−0.00, 0.00]	−0.00012 [−0.00038, 0.00013]	0.00008 [−0.00016, 0.00033]	**−0.00002 [−0.00002, −0.00001]**	−0.00 [−0.00001, 0.00]	−0.00025 [−0.0006, 0.00011]
** *σ* _2_ **	**0.109 [0.0794, 0.138]**	**−0.00039 [−0.0005, −0.00029]**	0.00484 [−0.0355, 0.0466]	**−0.0126 [−0.0198, −0.00544]**	0.0001 [−0.00006, 0.00026]	0.00002 [−0.00013, 0.00017]	**0.0232 [0.0126, 0.0337]**
** *σ* _3_ **	**0.992 [0.973, 1.01]**	**0.0023 [0.00223, 0.00236]**	−0.00546 [−0.0315, 0.0191]	0.00184 [−0.00276, 0.00654]	**−0.00017 [−0.00027, −0.00006]**	**0.00014 [0.00004, 0.00023]**	−0.00566 [−0.0126, 0.00103]
** *ε* _2_ **	**0.00622 [0.00153, 0.011]**	−0.00001 [−0.00011, 0.00009]	0.00202 [−0.00502, 0.0091]	−0.00013 [−0.00689, 0.00668]	−0.00001 [−0.00016, 0.00014]	−0.00001 [−0.00015, 0.00013]	−0.00206 [−0.012, 0.00793]
** *ε* _3_ **	−0.005 [−0.0107, 0.00066]	**−0.00035 [−0.00045, −0.00025]**	−0.00412 [−0.0123, 0.00411]	0.00118 [−0.00579, 0.00802]	0.00007 [−0.00008, 0.00023]	0.00003 [−0.00011, 0.00017]	0.00317 [−0.00701, 0.0134]
** *ψ* _2_ **	**0.108 [0.08, 0.136]**	**−0.00039 [−0.0005, −0.00029]**	0.00943 [−0.0282, 0.0498]	**−0.0126 [−0.0196, −0.00535]**	0.0001 [−0.00006, 0.00026]	0.00002 [−0.00013, 0.00016]	**0.0232 [0.0126, 0.0336]**
** *ψ* _3_ **	**10.0 [9.2, 10.8]**	**0.156 [0.152, 0.16]**	−0.191 [−1.22, 0.876]	**−0.282 [−0.555, −0.015]**	−0.00257 [−0.00853, 0.00336]	**0.0146 [0.00912, 0.0201]**	0.106 [−0.29, 0.514]

Note. Bolded values are statistically significant. ω_2_ = tonic volatility at level 2; ω_3_ = tonic volatility at level 3; β = inverse temperature parameter; μ_2_ = contingency beliefs; μ_3_ = volatility beliefs; σ_2_ = contingency uncertainty; σ_3_ = volatility uncertainty; ε_2_ = precision-weighted prediction error at level 2; ε_3_ = precision-weighted prediction error at level 3; ψ_2_ = precision weight at level 2; and ψ_3_ = precision weight at level 3; * = interaction term between variables; # = number. Reference groups: participants without an SI history and avoidance trials.

## Data Availability

The original data and code presented in the study are openly available at https://osf.io/3bk7e/overview?view_only=df26eabf9b35480483dcee0d163d8587 (accessed on 26 November 2025).

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
