# Peer review of "Computational Modeling of Uncertainty and Volatility Beliefs in Escape-Avoidance Learning: Comparing Individuals with and Without Suicidal Ideation"

_jpm, 2025, doi:10.3390/jpm15120604_

Round 1
Reviewer 1 Report
Comments and Suggestions for Authors
1-)please add more keywords if possible
2-)line 41, you may add updated reference.
3-)if possible please improve figure quality.
4-)you may add more sentences for the current study.
5-)please mention potential confounding factors in the limitations section.
6-)if chatbots are utilized to improve figures please mention it.
7-)please add more suggestions for further studies.
8-)please give more information about SONA systems. 9-)the following maybe unclear. why its related to the reference? At the conclusion of the 134 in-laboratory session, a suicide risk assessment was performed, and no participants 135 demonstrated elevated suicide risk as per Linehan et al. (2012). please be sure formulas are correct and relevant.Author Response
1. Please add more keywords if possible
Response: Thank you. We have expanded this to include the maximum number of keywords.
(p. 2): “Keywords: suicide; suicidal ideation; computational; belief; uncertainty; decision
making; cognitive; escape; reinforcement learning; hierarchical gaussian filter”
2. Line 41, you may add updated reference.
Response: The reference included is the paper where Ratcliff and McKoon establish the drift
diffusion model and its application to psychology. While this reference is critical to include, we
have included an additional, updated reference that discusses the incorporation of reinforcement
learning to drift diffusion.
(p. 4): “A limitation of these prior studies is that the drift diffusion approach utilized
assumes that the direction and rate of the decision-making process are solely driven by
the latent value of each choice given a particular stimulus (Pedersen et al., 2017; Ratcliff
& McKoon, 2008), without accounting for the uncertainty about these values.”
3. If possible please improve figure quality.
Response: Thank you. We have improved figure quality to match the journal’s standards
(1000dpi)
4. You may add more sentences for the current study.
Response: We have expanded the current section to discuss how this study builds on the
literature on reinforcement learning and suicidal ideation.
(p. 6): “This approach extends prior reinforcement learning research by directly modeling
how uncertainty and perceived environmental stability interact with escape and avoidance
processes in those with SI, addressing a key gap in prior work that has focused primarily
on value-based decision biases without considering uncertainty around them.”
5. Please mention potential confounding factors in the limitations section.
Response: We have expanded the limitation section to discuss potential confounding factors.
(p. 17): “Seventh, factors not included in the statistical analysis, such as baseline anxiety,
depression, and cognitive ability, may have influenced learning patterns and could
represent potential confounders.”
6. If chatbots are utilized to improve figures please mention it.
Response: Chatbots were not used to improve figures
7. Please add more suggestions for further studies.
Response: We expanded this section to add more future study suggestions.
(p. 19): “Future research should test whether these computational markers prospectively
predict the onset, maintenance, or escalation of suicidal thinking in high-risk populations,
and how they respond to specific treatment, thereby clarifying their potential as targets
for personalized intervention. Moreover, future studies should examine how belief-
updating dynamics relate to broader cognitive and emotional processes such as
regulation, flexibility, and impulsivity.”
8. Please give more information about SONA systems.
Response: We have expanded on the information about SONA.
(p. 6): “Participants were recruited through SONA Systems, a secure, web-based platform
used to recruit undergraduate participants for research studies completed in exchange for
course credit. This represented a convenience sample drawn from the University of Notre
Dame’s research participation pool.”
9. The following maybe unclear. why its related to the reference? At the conclusion of the 134 in-
laboratory session, a suicide risk assessment was performed, and no participants 135
demonstrated elevated suicide risk as per Linehan et al. (2012). please be sure formulas are
correct and relevant.
Response: The paper referenced informs the criteria for different levels of suicide risk. We have
revised this to make it clearer to the reader.
(p. 7): “Either a licensed clinical psychologist or a doctoral-level trained graduate
research assistant was available for risk management in the event of high suicide risk
(i.e., >4/7 intent to act on urges and an absence of a safety plan); no participants
demonstrated elevated suicide risk as per risk criteria suggested by Linehan et al.
(2012).”
Reviewer 2 Report
Comments and Suggestions for Authors
Dear Author(s),
Thank you for sending in your manuscript to the Journal of Personalized Medicine (JPM).
This paper dives into a fascinating area that blends computational psychiatry with suicide prevention. It aims to model how people, both with and without a history of suicidal ideation (SI), adjust their beliefs about action-outcome relationships and environmental unpredictability when trying to escape or avoid distressing situations. This topic holds significant theoretical and clinical value, potentially shedding light on the cognitive processes behind suicidal thoughts. I found your work quite engaging.
That said, the manuscript does need some revisions to enhance methodological transparency, interpretive accuracy, and overall clarity. Below, I’ve outlined my detailed comments, organized by section.
Abstract
1. Study design: Please clarify that this was a laboratory-based behavioral experiment utilizing a negative reinforcement learning task.
Introduction
2. Conceptual rationale: The way you differentiate between contingency, uncertainty, and volatility beliefs comes off as quite technical. It might be helpful to illustrate these concepts more clearly, perhaps with a brief explanatory figure.
3. Research gap: Make sure to explicitly state the gap this study is addressing, specifically, that previous models have focused on value-based decision biases but overlooked beliefs under uncertainty.
4. Hypotheses: The introduction could benefit from a clear and concise statement of your hypotheses. For instance, you might say that participants with a history of SI will show stronger contingency beliefs, less volatility updating, and greater uncertainty during escape trials.
Current study
5. I recommend renaming this section to “Study aims” for better clarity and to align with the standard structure.
Materials and Methods
6. Sampling Procedures: Could you please clarify how participants were recruited? For instance, was it through snowball sampling or purposive sampling?
7. Eligibility Criteria: This section currently mentions only high levels of depression as an exclusion criterion. Could you provide more details? Specifically: a) Was this determined through self-reporting or verified by a clinician? b) Were there any other exclusion criteria considered, such as psychiatric comorbidities, substance use, or neurological disorders, and how were these assessed? c) Were there specific inclusion criteria, like normal hearing due to the auditory aspect of the task, and how were these verified?
8. Sample Characteristics: The information provided in lines 130–134 includes descriptive details (like demographics) that would be better suited for the Results section. Please relocate this content accordingly.
9. Ethical Considerations: While the mention of adherence to the Declaration of Helsinki and IRB approval is good, it’s not enough. Please add a subsection at the end of the Methods that outlines procedures for ensuring anonymity, data storage, participant compensation, and risk management, especially given the sensitive nature of the study.
10. The Behavioral Task is described well, but it would be helpful to include a summary table or figure that clearly outlines the four trial types (active escape, passive escape, active avoidance, passive avoidance) along with their corresponding go/no-go mappings.
11. Please specify how performance feedback was given, how the intensity of the aversive sound was standardized across participants, and whether any participants chose to discontinue the task due to discomfort.
12. While the inclusion of equations in Computational Modeling is appreciated, this section is quite technical. It would be beneficial to simplify it for general readers by first summarizing the conceptual role of each HGF level (contingency, uncertainty, volatility) before diving into the mathematical details.
13. Lastly, please indicate which version of MATLAB and the HGF Toolbox was used.
14. Let’s make the analytic approach a bit clearer: a) Specify the priors that were used and whether the model assumptions (like normality and independence) were verified; b) Explain how any missing or invalid trials were dealt with; c) Share the reasoning behind modeling trial-by-trial trajectories instead of just focusing on aggregate parameters.
15. Don’t forget to mention the versions of the statistical software you used (like R or SPSS) and include reproducibility details in the Data Availability section (check the comment below for more on this).
Results
16. Participants' characteristics: As suggested, include in this section the summary of participant characteristics, including Table 1.
17. Table 1: It might be helpful to add effect sizes (like Cohen’s d or odds ratios) to enhance the p-values.
18. Main effects and interactions: Table 2 has a lot of information. Summarize the significant results in a brief paragraph that highlights the key parameters (like μâ‚‚, μ₃, σâ‚‚).
19. Interpretation: Make sure to clarify the direction of effects in simple terms. For example, you could say, “Participants with a history of SI grew more confident that escape responses would influence outcomes and viewed the environment as more stable.”
20. Consistency: Look out for minor formatting inconsistencies (like missing decimal points or irregular subscript formatting).
Discussion
21. Structure and focus: It would be beneficial to reorganize this section with a clear internal structure (headings aren’t necessary, but a good structure is recommended): Start with a summary of the main findings and how they relate to previous research; then interpret them within theoretical frameworks (like Escape Theory or Interpersonal Theory); discuss clinical and conceptual implications; and finally, address limitations and future research directions.
22. Interpretation: The connections you’re proposing between learning rates, uncertainty, and emotion regulation are fascinating but a bit speculative. Frame them as hypotheses or potential interpretations rather than definitive causal explanations.
23. Clinical relevance: The conversation around potential interventions, like repetitive skill practice to help stabilize coping models, is intriguing. However, it would be more appropriate to present this as a conceptual idea rather than as solid evidence-based recommendations.
24. Limitations: It’s also important to highlight any limitations related to the eligibility criteria (refer to previous comments). You should stress that an undergraduate sample might not capture the diverse range of suicidal experiences seen in clinical populations or among individuals with diagnosed mental health disorders.
25. Future directions: It would be worthwhile to investigate whether computational parameters can predict changes in suicidal ideation over time or responses to interventions, which aligns nicely with the journal’s focus on personalized medicine.
Conclusions
26. Let’s reiterate that the findings suggest, rather than prove, a connection between a history of suicidal ideation and specific patterns of belief updating.
27. You should be cautious not to overgeneralize the clinical implications, especially considering the non-clinical sample.
Language and Style
28. Make sure that the reference formatting and in-text citations adhere to MDPI style guidelines (including style, spacing, and punctuation).
Technical and Ethical Notes
29. The Data Availability statement needs to include the full OSF link or DOI to ensure reproducibility.
General Comment
This manuscript offers a significant contribution to the field of computational models of suicidal cognition, building on previous research by integrating belief uncertainty and volatility into the modeling framework. The study shows a solid methodology and theoretical foundation.
That said, the paper needs some revisions to improve methodological transparency, enhance interpretive clarity, and make it more accessible to readers from various disciplines. Addressing these points will bolster the manuscript’s scientific and clinical relevance.
Once revised, this study could add valuable insights to the literature on computational psychiatry and suicide research.
I hope these comments prove to be constructive and beneficial for the Author(s).
Best regards.
Author Response
Dear Author(s),
Thank you for sending in your manuscript to the Journal of Personalized Medicine (JPM).
This paper dives into a fascinating area that blends computational psychiatry with suicide prevention. It aims to model how people, both with and without a history of suicidal ideation (SI), adjust their beliefs about action-outcome relationships and environmental unpredictability when trying to escape or avoid distressing situations. This topic holds significant theoretical and clinical value, potentially shedding light on the cognitive processes behind suicidal thoughts. I found your work quite engaging.
That said, the manuscript does need some revisions to enhance methodological transparency, interpretive accuracy, and overall clarity. Below, I’ve outlined my detailed comments, organized by section.
Abstract
1. Study design: Please clarify that this was a laboratory-based behavioral experiment utilizing a negative reinforcement learning task.
Response: We have revised the abstract to add this context.
(p. 2): “Undergraduate students with (n=58) and without (n=62) lifetime SI history made active (Go) or passive (No-Go) choices in response to stimuli to escape or avoid an unpleasant state in a laboratory-based negative reinforcement task.”
Introduction
2. Conceptual rationale: The way you differentiate between contingency, uncertainty, and volatility beliefs comes off as quite technical. It might be helpful to illustrate these concepts more clearly, perhaps with a brief explanatory figure.
Response: Thank you. We have revised our introduction of these concepts (contingency beliefs, uncertainty, and volatility) to provide a clearer explanation and a more intuitive example illustrating how the computational parameters correspond to real-world cognitive processes.
(p.5): “Put simply, contingency beliefs reflect what individuals think will happen when they take a specific action; uncertainty reflects how confident they are in that expectation; and volatility reflects whether they believe the “rules of the environment” are stable or shifting. For example, within the Interpersonal Theory framework (Van Orden et al., 2010), a person with who has experienced SI and feels chronically disconnected might strongly believe that positive social interactions can relieve distress (a contingency belief) but be uncertain about whether such outcomes will actually occur (uncertainty), while also believing that their social circumstances rarely change (low perceived volatility). Together, the combination of strong yet uncertain contingency beliefs and low volatility expectations can reinforce hopelessness and rigid, escape-oriented thinking.”
3. Research gap: Make sure to explicitly state the gap this study is addressing, specifically, that previous models have focused on value-based decision biases but overlooked beliefs under uncertainty.
Response: We have extended the current study section to make the addressed gap clearer to the reader.
(p. 6): “This approach extends prior reinforcement learning research by directly modeling how uncertainty and perceived environmental stability interact with escape and avoidance processes in those with SI, addressing a key gap in prior work that has focused primarily on value-based decision biases without considering uncertainty around them.”
4. Hypotheses: The introduction could benefit from a clear and concise statement of your hypotheses. For instance, you might say that participants with a history of SI will show stronger contingency beliefs, less volatility updating, and greater uncertainty during escape trials.
Response: Thank you for your comment. Based on prior work, we had hypotheses regarding contingency beliefs, uncertainty, and volatility; however, our hypotheses regarding dynamic learning rates and precision weighted prediction errors remained exploratory.
(p. 6): “The present study employs an HGF to explore how individuals with and without a history of SI differ in their contingency beliefs, contingency uncertainty, and volatility beliefs during escape and avoidance learning. Building on prior work demonstrating escape biases in SI (Blacutt et al., 2025; Jaroszewski et al., 2025; Millner et al., 2019) and theoretical accounts of ambivalence and uncertainty during suicidal crises (Mitchell et al., 2024; Tsypes et al., 2022), we hypothesized that participants with SI histories would develop progressively stronger contingency beliefs across the task and maintain higher contingency uncertainty in escape trials compared to those without a history of SI. Given the novelty of dynamic learning rates and prediction weighted prediction errors in SI literature, analyses about these parameters remained exploratory. This approach extends prior reinforcement learning research by directly modeling how uncertainty and perceived environmental stability interact with escape and avoidance processes in those with SI, addressing a key gap in prior work that has focused primarily on value-based decision biases without considering uncertainty around them.”
Current study
5. I recommend renaming this section to “Study aims” for better clarity and to align with the standard structure.
Response: We have edited the title of this section.
Materials and Methods
6. Sampling Procedures: Could you please clarify how participants were recruited? For instance, was it through snowball sampling or purposive sampling?
Response: We have expanded the participant section to clarify our recruitment processes.
(p. 6): “Participants were recruited through SONA Systems, a secure, web-based platform used to recruit undergraduate participants for research studies completed in exchange for course credit. This represented a convenience sample drawn from the University of Notre Dame’s research participation pool.”
7. Eligibility Criteria: This section currently mentions only high levels of depression as an exclusion criterion. Could you provide more details? Specifically: a) Was this determined through self-reporting or verified by a clinician? b) Were there any other exclusion criteria considered, such as psychiatric comorbidities, substance use, or neurological disorders, and how were these assessed? c) Were there specific inclusion criteria, like normal hearing due to the auditory aspect of the task, and how were these verified?
Response: Thank you for your thoughtful questions. A) It was self-reported. B) These were the only exclusion criteria. C) There were no additional specific inclusion criteria. We have expanded the participant section to clarify these for the reader.
(p. 6-7): “Participants with and without a lifetime SI history were included; the only exclusion criteria were reporting high levels of current depressive symptomatology (i.e., score of 20 or greater on the Patient Health Questionnaire-9) (Kroenke et al., 2001) or current suicidality (i.e., reporting SI in the past week) due to ethical considerations, as the larger study included negative mood induction. These criteria were assessed via self-reported responses on the PHQ-9 prior to the laboratory session.”
8. Sample Characteristics: The information provided in lines 130–134 includes descriptive details (like demographics) that would be better suited for the Results section. Please relocate this content accordingly.
Response: This section has been moved to the results section.
(p. 11): “ Results
The final sample consisted of 120 undergraduate students from a private Midwestern university, who either had a history of SI (n = 58) or no history of SI (n = 62), and were invited to an in-laboratory session. Participants had a mean age of 19.1±1.3 years and self-identified as Asian (10.8%), Black (8.3%), Middle Eastern (4.2%), Multiracial (8.3%), and White (68.3%). See Table 1.”
9. Ethical Considerations: While the mention of adherence to the Declaration of Helsinki and IRB approval is good, it’s not enough. Please add a subsection at the end of the Methods that outlines procedures for ensuring anonymity, data storage, participant compensation, and risk management, especially given the sensitive nature of the study.
Response: Thank you for your thoughtfulness regarding the sensitive nature of the study. We have expanded the methods section to clarify our anonymity, data storage, compensation, and risk management processes.
(p. 7): “Data were de-identified via study-specific codes and were stored on an institutional secure internal server for sensitive data. Participants were compensated via SONA credit (1 credit/hour) and were assessed for suicide risk at the beginning and end of the laboratory session using the University of Washington Risk Assessment Protocol (Linehan et al., 2012). Either a licensed clinical psychologist or a doctoral-level trained graduate research assistant was available for risk management in the event of high suicide risk (i.e., >4/7 intent to act on urges and an absence of a safety plan); no participants demonstrated elevated suicide risk as per risk criteria suggested by Linehan et al. (2012). The study was carried out in accordance with the Declaration of Helsinki and was approved by the University of Notre Dame’s IRB.”
10. The Behavioral Task is described well, but it would be helpful to include a summary table or figure that clearly outlines the four trial types (active escape, passive escape, active avoidance, passive avoidance) along with their corresponding go/no-go mappings.
Response: Thank you for this suggestion. We have added a new table to clearly outline the response types.
(p. 26):
Table 1. Task conditions, types of responses, and escape/avoidance response types they map onto
Trial Description
Correct Response
Go
No-Go
Trial start: Aversive sound
Active escape
Passive escape
Correct response: Terminates sound
Incorrect response: Sound continues
Trial start: Silence
Correct response: Silence continues
Incorrect response: Sound starts
Active avoidance
Passive avoidance
11. Please specify how performance feedback was given, how the intensity of the aversive sound was standardized across participants, and whether any participants chose to discontinue the task due to discomfort.
Response: We have revised our description of the task to clarify the details of the feedback process.
(p. 8): “The aversive sound was the sound of a fork scraping on a slate, altered with a high-frequency sound, presented over headphones at 80-85dB. Each cue was presented for 1 second, during which participants could not respond, followed by a 2-second window where they could respond at any time. The aversive sound was presented for 2 seconds after incorrect choices (e.g., negative feedback), and silence was presented for 750ms after correct choices (e.g., positive feedback); after receiving the aversive sound or silence, a 1-second inter-trial interval with no sound and a screen displaying a white cross was presented. Before starting the task, participants received brief instructions stating that they would need to choose to either click or not click their mouse in response to images to either cease or prevent a loud and aversive sound. Further, to try to learn what the best response is for each image. Participants completed a total of 120 trials; no participant discontinued the experiment.”
12. While the inclusion of equations in Computational Modeling is appreciated, this section is quite technical. It would be beneficial to simplify it for general readers by first summarizing the conceptual role of each HGF level (contingency, uncertainty, volatility) before diving into the mathematical details.
Response: We have revised our discussion of the HGF and its parameters in the introduction to provide the reader with more clarity on what they mean. Further, we added a more intuitive example.
(p. 5): “Put simply, contingency beliefs reflect what individuals think will happen when they take a specific action; uncertainty reflects how confident they are in that expectation; and volatility reflects whether they believe the “rules of the environment” are stable or shifting. For example, within the Interpersonal Theory framework (Van Orden et al.,
2010), a person with SI who feels chronically disconnected might strongly believe that positive social interactions can relieve distress (a contingency belief) but be uncertain about whether such outcomes will actually occur (uncertainty) while also believing that their circumstances rarely change (low perceived volatility). Together, the combination of strong yet uncertain contingency beliefs and low volatility expectations can reinforce hopelessness and rigid, escape-oriented thinking.”
13. Lastly, please indicate which version of MATLAB and the HGF Toolbox was used.
Response: We have added this information.
(p. 9): “To estimate participants’ trial-by-trial beliefs, we fit the HGF to each individual’s task data separately using variational approximation, implemented in the HGF Toolbox version 7.1 (Frässle et al., 2021; Mathys et al., 2011, 2014) in MATLAB version 2025b with default initial parameters.”
14. Let’s make the analytic approach a bit clearer: a) Specify the priors that were used and whether the model assumptions (like normality and independence) were verified; b) Explain how any missing or invalid trials were dealt with; c) Share the reasoning behind modeling trial-by-trial trajectories instead of just focusing on aggregate parameters.
Response: We have revised the analytic section to make it clearer and transparent for the reader.
(p. 11-12): “All models used non-informative priors, and model assumptions were verified via visual inspection of residuals and random effects. Trial-by-trial trajectory modeling was employed to capture dynamic changes in belief updating across the task, as aggregate parameters would obscure temporal patterns of learning that may differ between groups with and without SI history. There was no missing trial-level data.”
15. Don’t forget to mention the versions of the statistical software you used (like R or SPSS) and include reproducibility details in the Data Availability section (check the comment below for more on this).
Response: Thank you for catching this. We have updated this throughout the manuscript to ensure versions are mentioned.
Results
16. Participants' characteristics: As suggested, include in this section the summary of participant characteristics, including Table 1.
Response: Thank you for the reminder. We have moved this section and the Table.
17. Table 1: It might be helpful to add effect sizes (like Cohen’s d or odds ratios) to enhance the p-values.
Response: Given the non-significance of most of the table, we did not include these throughout, but added to the result section to provide the standardized mean difference for age.
(p. 12): “Participants with a history of SI were significantly younger than those without an SI history (standardized mean difference = -0.64).”
18. Main effects and interactions: Table 2 has a lot of information. Summarize the significant results in a brief paragraph that highlights the key parameters (like μâ‚‚, μ₃, σâ‚‚).
Response: Thank you for this feedback. We have organized both the fixed and trial-level parameters to summarize results, organized by parameter type.
(p. 12-13): “Fixed Parameters
Bayesian mixed effects models did not reveal an effect of SI history, condition, or their interaction on perceptions of volatility and choice stochasticity (Table 3).
Trial-by-Trial Parameters
All trial-level parameters can be found in Table 3.
Contingency and volatility beliefs. We found a trial number by SI history interaction on contingency beliefs. Relative to those without SI experiences, participants with an SI history developed progressively stronger contingency beliefs across the task (Figure 2). Moreover, we found a negative interaction of trial number by SI history on volatility beliefs, suggesting that participants with an SI history became increasingly expectant that the task environment would remain unchanged.
Contingency and volatility uncertainty. We found an escape trial by SI history interaction on contingency uncertainty, suggesting that participants with an SI history maintained higher contingency uncertainty in escape trials compared to those without SI experiences, suggesting difficulty consolidating reward contingencies when actively terminating an aversive state (Figure 2). Further, we found a negative trial number by SI history interaction on volatility uncertainty, indicating that relative to those without an SI history, participants with an SI history became less uncertain about the (lack of) volatility throughout the task.
Precision Weighted Prediction Errors. We did not find any effect of SI history on contingency or volatility precision weighted prediction errors.
Dynamic Learning Rates. We found an escape trial by SI history interaction on contingency learning rate, which suggests that participants with an SI history weighted surprising outcomes (e.g., those with high prediction errors) more strongly when updating their contingency beliefs within that same trial than those without SI experiences, when actively trying to terminate the sound (Figure 2). We did not find any effect of SI history on volatility learning rate.”
19. Interpretation: Make sure to clarify the direction of effects in simple terms. For example, you could say, “Participants with a history of SI grew more confident that escape responses would influence outcomes and viewed the environment as more stable.”
Response: We have added a sentence to guide interpretation in sections that previously did not have one.
(p. 13): “Contingency and volatility uncertainty. We found an escape trial by SI history interaction on contingency uncertainty. Participants with an SI history maintained higher contingency uncertainty in escape trials compared to those without SI experiences, meaning they remained less confident about which actions would successfully terminate the sound. This suggests difficulty consolidating reward contingencies when actively terminating an aversive state (Figure 2). Further, we found a negative trial number by SI history interaction on volatility uncertainty, indicating that participants with an SI history became more certain that the environment would not change throughout the task.”
(p. 13): “Dynamic Learning Rates. We found an escape trial by SI history interaction on contingency learning rate, which suggests that participants with an SI history weighted surprising outcomes (e.g., those with high prediction errors) more strongly when updating their contingency beliefs within that same trial than those without SI experiences, when actively trying to terminate the sound (Figure 2). In other words, those with SI histories adjusted their expectations by a greater magnitude in response to unexpected results. We did not find any effect of SI history on volatility learning rate.”
20. Consistency: Look out for minor formatting inconsistencies (like missing decimal points or irregular subscript formatting).
Response: Thank you. We have addressed this throughout.
Discussion
21. Structure and focus: It would be beneficial to reorganize this section with a clear internal structure (headings aren’t necessary, but a good structure is recommended): Start with a summary of the main findings and how they relate to previous research; then interpret them
within theoretical frameworks (like Escape Theory or Interpersonal Theory); discuss clinical and conceptual implications; and finally, address limitations and future research directions.
Response: We wrote the discussion section to have a similar structure to what has been suggested, with a summary of main findings, followed by integration into theory and previous findings, stating clinical implications, and finishing with limitations. Future directions are discussed within the limitations and conclusion sections. We have added key sentences throughout the discussion to guide the reader through each section.
(p. 14): “These findings align with established cognitive and escape theories of SI (Baumeister, 1990; Dombrovski & Hallquist, 2022; Rudd, 2000)”
(p. 15): “In addition to being aligned with modern theories of suicide, these findings may provide a mechanistic account for empirical findings in related suicide literature. Al-Dajani et al. (2019) found that a stronger belief that suicide offers amplified the effect of low emotional clarity on SI at a six-month follow-up, suggesting that emotional uncertainty heightens the risk that escape-oriented beliefs sustain suicidal thinking. Our finding of elevated dynamic learning rates during escape trials in individuals with SI history suggests a potential computational mechanism underlying this relationship.”
(p. 16): “The computational insights from the present study yield several potentially important clinical implications for treating individuals with SI.”
(p. 17): “Several limitations warrant consideration when interpreting these findings.”
22. Interpretation: The connections you’re proposing between learning rates, uncertainty, and emotion regulation are fascinating but a bit speculative. Frame them as hypotheses or potential interpretations rather than definitive causal explanations.
Response: We have edited the language throughout the discussion and conclusion to soften our language and claims – ensuring the reader knows they are hypotheses/potential interpretations as opposed to casual explanations.
23. Clinical relevance: The conversation around potential interventions, like repetitive skill practice to help stabilize coping models, is intriguing. However, it would be more appropriate to present this as a conceptual idea rather than as solid evidence-based recommendations.
Response: We added a sentence at the end of the clinical relevance section to make it clear to the reader that these are conceptual/speculative.
(p. 17): “Given the non-clinical sample, these clinical implications should be considered preliminary theoretical applications that require empirical validation in clinical treatment settings before implementation.”
24. Limitations: It’s also important to highlight any limitations related to the eligibility criteria (refer to previous comments). You should stress that an undergraduate sample might not capture the diverse range of suicidal experiences seen in clinical populations or among individuals with diagnosed mental health disorders.
Response: We have expanded the limitation section to highlight the limitations of the undergraduate sample and broader eligibility criteria.
(p. 17): “Second, the sole recruitment of an undergraduate student sample with a relatively young age (mean 19.1 years) restricts our ability to determine whether these computational patterns persist across the lifespan. Third, exclusion of individuals with severe current depression or recent SI limits generalizability to clinical populations, as our sample may not capture the diverse range of suicidal experiences seen in individuals with active suicidality or diagnosed mental health disorders. Fourth, we did not exclude based on hearing impairments, potentially introducing variability in auditory perception of the aversive sound.”
25. Future directions: It would be worthwhile to investigate whether computational parameters can predict changes in suicidal ideation over time or responses to interventions, which aligns nicely with the journal’s focus on personalized medicine.
Response: Thank you. We expanded the future directions section to specifically include suggestions to examine longitudinal prediction and response to interventions.
(p. 19): “Future research should test whether these computational markers prospectively predict the onset, maintenance, or escalation of suicidal thinking in high-risk populations, and how they respond to specific treatment, thereby clarifying their potential as targets for personalized intervention.”
Conclusions
26. Let’s reiterate that the findings suggest, rather than prove, a connection between a history of suicidal ideation and specific patterns of belief updating.
Response: We have softened the language throughout the conclusion to make it clear that these are our interpretations and hypothetical connections.
(p. 19): “The present study advances understanding of SI by applying a computational lens that captures the dynamic interplay between belief formation, uncertainty processing, and escape motivation. The findings from the HGF model suggest that individuals with a history of SI develop increasingly rigid beliefs about environmental stability while simultaneously showing escape-specific persistence of contingency uncertainty, even as their uncertainty about volatility decreased across the task. These findings extend prior drift diffusion work by identifying not only an escape bias but also the impaired learning mechanisms that likely sustain it, potentially offering mechanistic insight into why suicidal thinking persists despite repeated ineffective escape attempts. These results suggest it may be prudent for clinicians treating individuals with SI to emphasize strengthening precise, reliable coping strategies and collaboratively calibrating expectations about the effectiveness of these techniques. Future research should test whether these computational markers prospectively predict the onset, maintenance, or escalation of suicidal thinking in high-risk populations, and how they respond to specific treatment, thereby clarifying their potential as targets for personalized intervention. Moreover, future studies should examine how belief-updating dynamics relate to broader cognitive and emotional processes such as regulation, flexibility, and impulsivity.”
27. You should be cautious not to overgeneralize the clinical implications, especially considering the non-clinical sample.
Response: Thanks for this suggestion. We have added a sentence at the end of the clinical implication paragraph to make this clear to readers.
(p. 17): “Given the non-clinical sample, these clinical implications should be considered preliminary theoretical applications that require empirical validation in clinical treatment settings before implementation.”
Language and Style
28. Make sure that the reference formatting and in-text citations adhere to MDPI style guidelines (including style, spacing, and punctuation).
Response: We have made sure the MDPI style guidelines are met throughout.
Technical and Ethical Notes
29. The Data Availability statement needs to include the full OSF link or DOI to ensure reproducibility.
Response: We have added the OSF link.
(p. 20): “Data Availability Statement
All data and code to reproduce the study are available on the corresponding author’s Open Science Framework: https://osf.io/3bk7e/overview?view_only=df26eabf9b35480483dcee0d163d8587”
General Comment
This manuscript offers a significant contribution to the field of computational models of suicidal cognition, building on previous research by integrating belief uncertainty and volatility into the modeling framework. The study shows a solid methodology and theoretical foundation.
That said, the paper needs some revisions to improve methodological transparency, enhance interpretive clarity, and make it more accessible to readers from various disciplines. Addressing these points will bolster the manuscript’s scientific and clinical relevance.
Once revised, this study could add valuable insights to the literature on computational psychiatry and suicide research.
I hope these comments prove to be constructive and beneficial for the Author(s).
Best regards.
Round 2
Reviewer 1 Report
Comments and Suggestions for Authors
thank you very much for the revised file.
I accept the paper.
Author Response
We thank Reviewer 1 for their time and effort reviewing the manuscript.

Reviewer 2 Report
Comments and Suggestions for Authors
Dear Author(s),
Thank you for submitting the revised version of your manuscript.
I have read your responses and examined the updated manuscript.
I appreciate the way in which you addressed the previous comments. The revision has improved the manuscript in terms of clarity of design, methodological transparency, and overall structure. The description is now more accessible, and the Discussion is better aligned with the study objectives and the exploratory nature of the analyses.
All previous comments and suggestions have been fully addressed, and the revised manuscript is now clear. I have no further revisions to request.
I wish you the best with your research.
Best regards.
Author Response
We thank Reviewer 2 for their detailed review and for acknowledging the
improvements made in the revised manuscript.